# Elucidating the Associated Biological Function and Clinical Significance of *RHOJ* Expression in Urothelial Carcinoma

**DOI:** 10.3390/ijms241814081

**Published:** 2023-09-14

**Authors:** Xin-Jie Lu, Hsing-Fan Lai, Sheng-Cheng Wu, Chin-Li Chen, Yi-Lin Chiu

**Affiliations:** 1Department of Biochemistry, National Defense Medical Center, Taipei 114, Taiwan; xinjielu@mail.ndmctsgh.edu.tw (X.-J.L.); michael199622@gmail.com (H.-F.L.); yilin1107@mail.ndmctsgh.edu.tw (Y.-L.C.); 2Graduate Institute of Life Sciences, National Defense Medical Center, Taipei 114, Taiwan; 3Division of Hematology and Oncology, Department of Internal Medicine, Tri-Service General Hospital Penghu Branch, Magong 880, Taiwan; 4Division of Urology, Department of Surgery, Tri-Service General Hospital, National Defense Medical Center, Taipei 114, Taiwan

**Keywords:** *RHOJ* expression in urothelial carcinoma, tumor microenvironment, prognostic biomarkers in bladder cancer

## Abstract

Urothelial cancer, a common urinary system malignancy, often presents treatment challenges due to metastasis and chemotherapy side effects. Angiogenesis, crucial for tumor growth, has become a target for drug development. This study explores the expression, prognostic value, and clinical correlation of *RHOJ* in the TCGA BLCA, GSE31684, and GSE32894 datasets. We identify common differentially expressed genes across these databases and utilize g:Profiler and Cytoscape ClueGO for functional assessment. Further, we perform a gene set enrichment analysis (GSEA) using Hallmark gene sets and use the imsig package for immune cell infiltration analysis. Our analysis indicates that *RHOJ* expression levels significantly impact survival rates, tumor progression, and immune response in urothelial tumors. High *RHOJ* expression correlated with poor prognosis, advanced disease stages, and an increase in monocyte population within the tumor microenvironment. This aligns with current literature indicating a key role of immune infiltration in bladder cancer progression and treatment response. Moreover, the GSEA and imsig results further suggest a potential mechanistic link between *RHOJ* expression and immune-related pathways. Considering the increasing emphasis on immunotherapeutic strategies in bladder cancer management, our findings on *RHOJ*’s potential as a diagnostic biomarker and its association with immune response open new avenues for therapeutic interventions.

## 1. Introduction

Urothelial tumors, the most common malignant tumors of the urinary system, consist of epithelial cell cancers in different parts of the urinary tract, such as the renal pelvis, ureter, and bladder. They account for about 90% of urological cancers in the United States and are the sixth most common cancer generally. Clinically, approximately 75% of bladder cancers are non-muscle-invasive bladder cancer (NMIBC), and the remaining 25% are muscle-invasive bladder cancer (MIBC). Although most cases are initially diagnosed as NMIBC, up to 50% of patients may experience recurrence within five years following treatment. Of these recurrences, around 10–20% will progress to more aggressive muscle-invasive or metastatic urothelial cancer, resulting in the development of new lesions [1]. The overall five-year survival rate for urothelial tumors is approximately 77%. However, in cases with regional or distant metastasis, the five-year mean survival rate drastically declines to between 8 and 39% [2]. This emphasizes the substantial obstacle presented by metastasis in urothelial tumor clinical management.

Chemotherapeutic agents such as Cisplatin, Doxorubicin, and Cyclophosphamide are commonly administered for the treatment of metastatic urothelial tumors. These agents can be administered either singly or in combination and suppress the growth of urothelial cancer at an early stage. The combination of Methotrexate, Vinblastine, Doxorubicin, and Cisplatin (MVAC), for instance, has been shown to significantly increase the average survival time in patients with metastatic urothelial tumors in comparison to the use of single chemotherapy agents [3]. However, the toxicity and side effects of administering multiple chemotherapy drugs frequently render it arduous for numerous elderly patients to withstand continuous treatment. Moreover, throughout the treatment process, these patients may also necessitate prolonged hospital stays to oversee the onset of neutropenic sepsis [4]. Thus, developing a combination of localized resection surgery for early detection of NMIBC, along with preventative drug therapy to reduce metastasis, could potentially minimize patient distress and improve overall survival rates.

Tumor angiogenesis is a critical necessity for tumor development [5,6]. Signaling pathways related to angiogenesis become activated during tumor growth [6,7]. Furthermore, the vascular system of the tumor plays a significant role as one of the crucial pathways for tumor cells to metastasize to distant organs [8]. In the last decade, this hypothesis has encouraged the creation of several angiogenesis-inhibiting agents (AIAs) [9,10,11]. Many medicines are designed to inhibit the vascular endothelial growth factor (VEGF) or its receptors, and their effectiveness has been widely reported [7,12]. In addition, there is a significant interest in developing medications that suppress other pathways related to angiogenesis [11,13,14,15,16]. The existing AIAs mainly operate by obstructing neovascularization to curb tumor expansion. Rather than inducing tumor regression, these drugs usually hinder tumor expansion [9,17]. In certain instances, despite receiving AIA treatment, a substantial vascular network may persist in the tumor [17]. Preclinical research has also indicated that tumor cells could undergo a phenotypic transformation towards a more aggressive phenotype, resulting in an increased invasive and metastatic potential following AIA therapy [18,19]. In addition to enhancing AIA effectiveness, the adverse effects remain a significant concern. Since the VEGF and its receptors are expressed in both normal tissues and tumors, the present usage of AIAs often results in unwanted reactions like hypertension, proteinuria, and bleeding [20,21]. Identifying definitive biomarkers to distinguish tumor vasculature from normal would significantly advance the development of more specific and effective targeted strategies.

Rho GTPases have recently emerged as regulators of vascular morphogenesis and homeostasis [22]. Mechanistically, Rho GTPases are recognized as essential downstream targets of VEGF signaling in endothelial cells (ECs). They play a crucial role in maintaining an optimal balance between different types of Rho GTPases during the process of angiogenesis, which includes EC migration, EC proliferation, extracellular matrix degradation, vascular morphogenesis, and vascular integrity [22,23]. The RhoJ protein, encoded by the *RHOJ* gene, is mainly expressed as a Rho GTPase in ECs [24,25,26,27]. Its expression is regulated by the endothelial transcription factor ERG. Recent research has shown that RhoJ is a key regulator of EC migration and tubulogenesis [26,28]. During embryonic development, RhoJ has specific expression in the murine dorsal aorta and interstitial vessels, as well as postnatal retinal vessels [24,25]. RhoJ-deficient mice exhibit stunted growth and a higher rate of regression in retinal blood vessels during postnatal development [26]. These findings emphasized the vital role of RhoJ in sustaining a balance between vascular formation and regression, ultimately affecting vascular remodeling. Nevertheless, the expression and function of RhoJ in tumor angiogenesis remain uncertain.

This study analyzes the expression of *RHOJ* in patients with urothelial tumors employing the TCGA BLCA, GSE31684, and GSE32894 databases. The investigation aims to correlate the expression of *RHOJ* with survival rates, explore the effect of *RHOJ* expression on overall gene expression in tumor cells, scrutinize the characteristics of the tumor microenvironment, and analyze the relationship between *RHOJ* and immune cell infiltration in clinical samples of urothelial carcinoma.

## 2. Results

### 2.1. Correlation of RHOJ Expression and Prognosis in Patients with Urothelial Tumors

We analyzed the relationship between *RHOJ* expression and prognosis in patients with urothelial tumors using the TCGA BLCA (The Cancer Genome Atlas Urothelial/Bladder Carcinoma), GSE31684, and GSE32894 databases. The bladder cancer patients were stratified into two groups based on their *RHOJ* expression levels: high expression and low expression. We then employed a Kaplan–Meier survival analysis to generate survival curves for these two groups and to assess the differences in prognosis. In analyzing the three databases, a significant disparity in survival emerged between the group expressing high levels of *RHOJ* and those with low expression levels. This highlights a poorer prognosis and significantly lower survival rates in bladder cancer patients with elevated *RHOJ* expression (Figure 1). Our findings suggest that *RHOJ* may serve as a promising prognostic indicator with noteworthy clinical implications for the diagnosis and treatment of urothelial tumors.

Intravesical Bacillus Calmette–Guérin (BCG) therapy is a common postoperative treatment following the transurethral resection of bladder tumors, potentially offering better outcomes. In this study, we aimed to assess the expression of *RHOJ* in bladder cancer and its implications for prognosis in patients treated with BCG. We utilized data from the TCGA BLCA database, which recorded 58 bladder cancer patients, with 34 receiving BCG and the remaining 24 undergoing transurethral resection alone. *RHOJ* expression levels were examined and patients were categorized into *RHOJ* high and *RHOJ* low groups based on their expression profiles. There was no significant difference in *RHOJ* expression between patients who did and did not receive BCG (Wilcoxon *t*-test *p*: 0.43) (Figure 1B). However, when categorized by *RHOJ* expression, a noticeable pattern emerged (Figure 1C). In the *RHOJ* low group, patients who received BCG showcased a better prognosis compared to those who did not. Intriguingly, no significant survival difference was observed in the *RHOJ* high group between the two treatments. Specifically, *RHOJ* low expression might serve as a predictor for better outcomes post BCG, but such advantages seem to diminish in patients with high *RHOJ* expression. This positions *RHOJ* as a potential biomarker for assessing the prognosis of BCG-treated bladder cancer patients.

### 2.2. Association of RHOJ Gene Expression and Promoter Methylation Level with Bladder Cancer Metastasis and Clinical Features

To investigate possible links between *RHOJ* gene expression and cancer cell metastasis in individuals with bladder cancer, we utilized the MEXPRESS tool to analyze clinical data obtained from the TCGA BLCA database. These clinical data covered various factors including patient age and gender, survival rate, tumor diagnosis category, pathological stage, TNM stage, and the presence of lymphovascular invasion. From this analysis, a clear link between *RHOJ* gene expression and lympho-vascular invasion, tumor stage, metastasis stage, lymph node invasion, and pathological stage emerged (Figure 2). This indicates that *RHOJ* gene expression levels could possibly serve as a valuable biomarker for the clinical diagnosis of metastatic tendencies in bladder or urothelial tumors. Notably, a significant negative correlation was found between the degree of methylation in the promoter region of the *RHOJ* gene and its expression (r = −0.424 and −0.355). This suggests that *RHOJ* expression may be regulated epigenetically.

### 2.3. Gene Expression Analysis in Urothelial Tumors with High RHOJ Expression

To obtain additional insights into the impact of elevated *RHOJ* expression on gene expression in patients with urothelial tumors, Volcano plot analyses were conducted on the TCGA BLCA, GSE31684, and GSE32894 databases (Figure 3). The outcomes disclosed that 1839 genes exhibited increased expression, while 480 genes exhibited declined expression in the TCGA BLCA database. In the GSE31684 database, 981 genes showed elevated expression, whereas 621 genes demonstrated decreased expression. Lastly, the GSE32894 database showed that 134 genes displayed increased expression, while 25 genes displayed decreased expression. A Venn Diagram analysis was conducted to identify similarities among these genes. The analysis revealed that 89 genes exhibited an increasing trend in all three databases. These findings imply a close potential association between high *RHOJ* expression and the upregulation of these genes.

### 2.4. Functional Analysis of RHOJ-Associated Genes in Urothelial Tumors with High RHOJ Expression

Through an analysis of the TCGA BLCA, GSE31684, and GSE32894 databases, we identified 89 genes that demonstrated an increased expression trend in patients with urothelial tumors and high *RHOJ* expression. For additional insights into the functional implications of these genes, we conducted further analysis using the g:Profiler tool (Figure 4). From the Gene Ontology (GO) analysis, it appears that these genes may be involved in modifying the tumor microenvironment. This is suggested by the significant enrichment in extracellular matrix (ECM) constituents, collagen binding, and ECM organization, which are both related to molecular function and biological process. Changes in the ECM are often associated with tumor growth, invasion, and metastasis. The terms related to development processes and anatomical structure development might hint at the genes’ roles in tumor progression and metastasis. The analysis of the cellular component (CC) indicates the involvement of extracellular matrix and extracellular space, highlighting the probable participation of these genes in ECM remodeling. Regarding pathway analysis, the KEGG and REACTOME databases exhibit enrichment in processes connected with protein digestion and absorption, vascular smooth muscle contraction, focal adhesion, and leukocyte *trans*-endothelial migration. These processes have been associated with tumor growth, metastasis, and angiogenesis, indicating that the genes may be involved in these essential aspects of cancer progression. The involvement of WikiPathways (WP) and transcription factors (TF) in burn wound healing, as well as miRNA targets in ECM and membrane receptors, has been found [28,29]. The association with wound healing is noteworthy, as cancer is frequently likened to a “wound that never heals,” indicating that these genes may contribute to sustaining a pro-tumorigenic milieu. MicroRNAs (miR-29b-3p, miR-29a-3p, and miR-27a-5p) have been linked to these genes, hinting at potential post-transcriptional regulation [30,31]. The Human Phenotype Ontology (HPO) provides a standardized vocabulary for describing phenotypic abnormalities observed in human diseases. The linkage of “Abnormality of bladder morphology” and “Abnormal cerebral artery morphology” in the HPO suggests a potential genetic or molecular association between these two phenotypes. Murugapoopathy and Gupta et al. extensively discussed the congenital anomalies of the kidneys and urinary tracts (CAKUT) [32]. These anomalies encompass a wide spectrum of structural malformations that arise from defects in the embryologic development of the urinary system. The fact that abnormalities in bladder morphology are part of the CAKUT spectrum might hint at developmental or genetic disturbances that could simultaneously impact both the urinary and cerebrovascular systems. On a related note, Leoni et al. shed light on the prevalence of bladder cancer in patients with Costello syndrome [33]. Costello syndrome, a rare genetic disorder, has a spectrum of phenotypic manifestations. While their focus is on the higher predisposition of bladder cancer in these patients, it is pivotal to consider the broader genetic alterations associated with such syndromes. The underlying genetic mutations could concurrently influence multiple organ systems, potentially explaining the diverse clinical manifestations.

### 2.5. Clustering and Functional Analysis of the 89 RHOJ-Associated Genes

Based on the clustering analysis and functional assessment using Cytoscape and ClueGO, it was found that the three most significant gene clusters were linked to important biological processes (Figure 5). The most substantial cluster, which constituted 52.1% of the genes, was mainly related to extracellular matrix degradation. This suggests that a considerable number of the genes may have a hand in tissue remodeling, which could, in turn, lead to the development of various cellular behaviors such as migration, invasion, and adhesion. In the context of cancer, these processes are crucial, as they may result in metastasis. The second most significant cluster, composing 33.61% of the genes, exhibited connections with Small Leucine-rich Proteoglycans (SLRPs) binding to TGF-beta. This implies a function in the regulation of growth factor activity and the influence on cellular proliferation, differentiation, and apoptosis. SLRPs and TGF-beta are renowned for their critical roles in several pathologies, like fibrosis and cancer. The third largest cluster, which constitutes 5.88% of the genes, was linked to the activation of PAKs by RHO GTPases. RHO GTPases, for instance *RHOJ*, serve as crucial controllers of cell motility and can also contribute to cancer progression by encouraging cell migration and invasion. The PAKs, or p21-activated kinases, are effector molecules downstream of RHO GTPases that have the capacity to alter cytoskeletal restructuring and cell motility.

In summary, the Gene Ontology and pathway analyses conducted via Cytoscape and ClueGO show a potential correlation with processes related to tissue remodeling, growth factor regulation, and cell motility, indicating a plausible mechanism in which upregulating these genes may contribute to cancer progression, specifically facilitating invasive and migratory behaviors.

### 2.6. Association of High RHOJ Expression on Hallmark Gene Sets in Urothelial Tumors

To further investigate the impact of high *RHOJ* expression on gene sets in urothelial tumors, we conducted a gene set enrichment analysis (GSEA) (Figure 6). The enrichment analysis of Hallmark gene sets in the TCGA BLCA, GSE31684, and GSE32894 databases revealed a positive enrichment of gene sets under conditions of high *RHOJ* expression. Consistent gene sets enriched in the Hallmark pathways included epithelial–mesenchymal transition (FDR < 0.001, NES = 2.349 in TCGA BLCA; FDR < 0.001, NES = 2.824 in GSE31684; FDR < 0.001, NES = 2.539 in GSE32894), allograft rejection (FDR < 0.001, NES = 2.053 in TCGA BLCA; FDR < 0.001, NES = 2.501 in GSE31684; FDR < 0.001, NES = 2.255 in GSE32894), TNF alpha signaling via NFKB (FDR < 0.001, NES = 2.014 in TCGA BLCA; FDR < 0.001, NES = 2.218 in GSE31684; FDR < 0.001, NES = 2.078 in GSE32894), and inflammatory response (FDR < 0.001, NES = 1.996 in TCGA BLCA; FDR < 0.001, NES = 2.221 in GSE31684; FDR < 0.001, NES = 2.318 in GSE32894). These gene sets are closely linked to the immune response. Furthermore, it was discovered that five gene sets in the TCGA BLCA database, seven gene sets in the GSE31684 database, and seven gene sets in the GSE32894 database are linked to the immune response. This implies that urothelial tumors with high *RHOJ* expression may display an increased immune response in the tumor microenvironment.

On the other hand, there were also negatively enriched gene sets within the Hallmark gene sets in these databases. Consistent gene sets included E2F targets (FDR < 0.001, NES = −2.291 in TCGA BLCA; FDR < 0.001, NES = −1.158 in GSE31684; FDR < 0.001, NES = −1.622 in GSE32894), MYC targets v1 (FDR < 0.001, NES = −2.138 in TCGA BLCA; FDR = 0.14, NES = −1.174 in GSE31684; FDR < 0.001, NES = −2.116 in GSE32894), MYC targets v2 (FDR < 0.001, NES = −2.015 in TCGA BLCA; FDR = 0.082, NES = −1.309 in GSE31684; FDR < 0.001, NES = −2.009 in GSE32894), and the G2M checkpoint (FDR < 0.001, NES = −1.665 in TCGA BLCA; FDR = 0.037, NES = −1.286 in GSE31684; FDR < 0.001, NES = −1.543 in GSE32894). These gene sets are associated with cell division, indicating that urothelial tumors with high *RHOJ* expression may exhibit a lower degree of cellular proliferation.

### 2.7. Association between RHOJ Expression and Tumor Immune Microenvironment in Urothelial Tumors

Finally, we investigated the association between *RHOJ* expression and the tumor immune microenvironment in urothelial tumors. Initially, the ImSig tool was used to analyze the TCGA BLCA and GSE31684 databases, evaluating the correlation between *RHOJ* expression and multiple immune cell populations (Figure 7). In the TCGA BLCA database, high *RHOJ* expression showed a positive correlation with macrophages (r = 0.21), monocytes (r = 0.20), natural killer cells (r = 0.14), and T cells (r = 0.28), while displaying a negative correlation with tumor cell proliferation (r = −0.13). In the GSE31684 database, a strong correlation with monocytes (r = 0.43) was observed with high *RHOJ* expression. These findings indicate that urothelial cancer patients with high *RHOJ* expression may possess a significant amount of monocytes within the tumor microenvironment. To explore further the correlation between *RHOJ* and the immune microenvironment, we employed the immune suppressive signature sets from the Tumor Immune Dysfunction and Exclusion (TIDE) webtool [34]. Using upregulated and downregulated gene sets, we generated integrated scores using the singscore method to evaluate the association with *RHOJ* expression [35]. The results showed a significantly positive correlation between *RHOJ* expression and both regulatory T cells (Treg) and T cell exhaustion (Texhaust) (TCGA BLCA: Treg r = 0.46, Texhaust r = 0.47; GSE31684: Treg r = 0.25, Texhaust r = 0.28) (Figure 8). Similar findings were identified in the GSE32894 database, as shown in the Appendix A. These results offer insights into the correlation between increased *RHOJ* expression and immune cell suppression in urothelial tumors. They emphasize the importance of the tumor immune microenvironment in urothelial cancer.

## 3. Discussion

The identification of novel prognostic markers and potential therapeutic targets is crucial for enhancing treatment strategies and outcomes in bladder cancer, one of the most prevalent types of cancer globally. This study investigates the function and consequences of *RHOJ*, a member of the Rho GTPase family, in patients with bladder cancer. Notably, our results suggest that *RHOJ* could have a notable influence on bladder cancer, particularly regarding disease progression and survival rates. We discovered an association between elevated *RHOJ* expression in bladder cancer patients and inferior prognosis with decreased survival rates, indicating its potential value as a prognostic indicator. These results are consistent with previous investigations where amplified *RHOJ* expression was connected to detrimental clinical consequences in distinct cancer forms [36,37,38,39,40]. Moreover, the high expression of *RHOJ* demonstrated a significant correlation with diverse clinical features linked to cancer metastasis and disease advancement, highlighting its worth as a crucial biomarker for disease severity and tumor aggression. The alignment of *RHOJ* expression with metastasis corresponds to its established function in cell migration and angiogenesis in endothelial cells. *RHOJ* has been reported to regulate cytoskeletal dynamics, which is a crucial process in cell migration and invasion, thereby contributing to the metastatic potential of cancer cells [36,37,39,41,42]. The proteins encoded by the Rho family of genes, which includes *RHOJ*, are involved in the regulation of the actin cytoskeleton in cells, influencing cell structure, division, and migration. In the context of tumor biology, certain members of the Rho family have been implicated in various processes, including neoangiogenesis, which is the formation of new blood vessels from the existing vasculature, a crucial process for tumor growth and metastasis. Ischemia, or insufficient blood supply to tissues, can induce neoangiogenesis or angiogenesis as a compensatory mechanism to restore oxygen and nutrient supply. Rho proteins, including *RHOJ*, can potentially play a role in this process. *RHOJ*, in particular, is a known player in the endothelial cell cytoskeleton reorganization, a crucial aspect of neoangiogenesis. Recent studies have shown that *RHOJ* is predominantly expressed in endothelial cells and plays a significant role in regulating their migration, a crucial step in angiogenesis. Furthermore, *RHOJ* is believed to be activated under ischemic conditions, suggesting its potential role in ischemia-induced angiogenesis. However, the precise mechanisms underlying this association in bladder cancer remain to be determined.

Our analysis showed a cluster of genes that had been consistently upregulated across multiple datasets in patients with high *RHOJ* expression. These genes were primarily linked with the organization of extracellular matrix and binding of platelet-derived growth factor, suggesting a probable mechanism through which *RHOJ* could affect the tumor microenvironment and carcinogenesis. Furthermore, the analysis of gene sets for enrichment provided interesting insights into the potential functional effects of *RHOJ* expression in bladder cancer. We have observed a positive enrichment of gene sets linked with epithelial–mesenchymal transition, allograft rejection, and inflammatory response, recognized to play important roles in cancer progression and immune evasion. This implies that *RHOJ* could play a role in regulating tumor–immune system interactions, thereby possibly contributing to immune evasion, a trait frequently observed in aggressive cancers. The survival analysis in TCGA BLCA reveals that in bladder cancer samples with low *RHOJ* expression, patients who received BCG treatment had a better prognosis. In contrast, samples with high *RHOJ* expression did not show any advantage from BCG treatment. This observation suggests that immune suppression associated with *RHOJ* expression might be antagonistic to the adaptive immunity driven by BCG treatment [43,44].

Notably, we also discovered a depletion in gene sets linked to cell division, such as E2F targets and MYC targets, hinting at a potential inhibitory influence of *RHOJ* on cell proliferation. This paradoxical observation requires further investigation, given that *RHOJ* has previously been linked to promoting cell proliferation and survival in other forms of cancer [36,37,38,39]. The variation in *RHOJ* expression across different immune cell populations indicates its potential impact on the tumor immune microenvironment, highlighting its significance in cancer immunotherapy. Notably, patients undergoing radical cystectomy for bladder cancer with a high monocyte to lymphocyte ratio are likely to have a poorer prognosis [45,46]. Overexpression of *RHOJ* can elevate the mRNA and protein levels of the pro-inflammatory vascular cell adhesion molecule 1 (VCAM-1) and intercellular adhesion molecule 1 (ICAM-1) in endothelial cells (ECs). This, in turn, leads to an escalation in the adhesion of monocytes to ECs and tumor tissue [47]. Therefore, the high expression of *RHOJ* in urothelial cancer may be related to an increased monocyte to lymphocyte ratio and potentially affect resistance to immunotherapy [48]. However, the molecular mechanisms behind this require further investigation. With the recent advances in immune checkpoint inhibitors for the treatment of bladder cancer, comprehending the role of *RHOJ* in immune modulation could yield valuable insights for optimizing immunotherapy strategies.

In the realm of therapeutic interventions for tumors, BCG treatment holds a pivotal role, especially concerning its mechanism of inducing adaptive immunity. Our recent analyses have shed light on a potentially valuable clinical tool: staining tumors for *RHOJ* expression. The *RHOJ* staining strategy, derived from our post-BCG treatment analyses, appears to be promising in facilitating a swift determination of a patient’s response to BCG-mediated adaptive immune reactions. Understanding a patient’s response to treatment is paramount, not just for gauging the immediate therapeutic impact, but also for predicting the long-term prognosis. The presence or absence of *RHOJ* expression could potentially serve as a biomarker, guiding clinicians in making more informed decisions about future interventions or adjustments to the current treatment regimen.

Despite these promising findings, it is important to acknowledge several limitations. Our study is primarily based on an in silico analysis, which, though effective for hypothesis generation, needs experimental validation. Subsequent studies should also investigate the molecular mechanisms that connect *RHOJ* expression to the observed clinical and molecular phenotypes, which would bolster the case for *RHOJ* as a potential therapeutic target.

## 4. Materials and Methods

### 4.1. Data Availability

The datasets analyzed in this study were obtained from The Cancer Genome Atlas (TCGA BLCA) database and two publicly available Gene Expression Omnibus (GEO) databases, namely GSE31684 and GSE32894 [49,50,51]. TCGA BLCA is an open-source database that contains multi-dimensional maps of key genomic changes in 33 types of cancer, including bladder urothelial carcinoma (BLCA). The database contains data on clinical characteristics, gene expression, and survival outcomes, among others, which can be accessed at https://portal.gdc.cancer.gov/ (accessed on 11 November 2022) GSE31684 includes microarray data from 93 bladder cancer patients who underwent radical cystectomy to determine gene expression patterns associated with clinical and prognostic variables. This dataset can be accessed at https://www.ncbi.nlm.nih.gov/geo/query/acc.cgi?acc=GSE31684 (accessed on 16 November 2022) GSE32894 includes gene expression profiles from 308 urothelial carcinomas. The gene expression profiles were obtained using the Illumina HumanHT-12 V3.0 expression beadchip arrays at the SCIBLU Genomics Centre at Lund University, Sweden. The dataset can be accessed at https://www.ncbi.nlm.nih.gov/geo/query/acc.cgi?acc=GSE32894 (accessed on 16 November 2022). Please refer to Supplementary Appendix A for the list of differentially expressed genes in each database according to the RHOJ expression classification.

### 4.2. Survival Analysis

We applied R survival (v3.5-5) and survminer (v0.4.9) packages to conduct a Kaplan–Meier survival analysis on *RHOJ* expression data. Within the TCGA and NCBI GEO datasets, the collection and utilization of data were in compliance with the respective policies and guidelines. Kaplan–Meier survival analysis and log-rank tests were conducted to compare survival rates between two groups, one with high *RHOJ* expression and the other with low *RHOJ* expression. Hazard ratios (HR) with 95% confidence intervals (CI) and *p*-values were calculated using univariate Cox proportional hazards regression. Differences in survival were visualized through Kaplan–Meier curves. All analyses were carried out using R software (v4.2.1).

### 4.3. MEXPRESS Analysis

MEXPRESS (http://mexpress.be (accessed on 11 November 2022)) was accessed, which is a web tool designed to provide easy visualization of TCGA expression, DNA methylation, and clinical data, as well as the relationships among them [52]. The BLCA dataset was selected from the drop-down list of available TCGA datasets. ‘*RHOJ*’ was entered into the search bar to bring up related results. The associations between *RHOJ* gene expression, methylation status, and clinical features relevant to cancer metastasis and prognosis were examined using the provided visualization and statistical tools.

### 4.4. Functional Enrichment Analysis and Visualization

The g profiler tool was employed, which can be accessed at https://biit.cs.ut.ee/gprofiler/gost (accessed on 12 November 2022) [53]. Functional profiling of the 89 differentially expressed genes was performed using g:GOST tool. The analysis was conducted using the default parameters. Selected data sources for this analysis included Gene Ontology (GO) categories molecular function, cellular component, and biological process, along with KEGG, Reactome (REAC), WikiPathways (WP), TRANSFAC (TF), miRTarBase (MIRNA), and the Human Phenotype Ontology (HP). The results from each dataset were filtered using a false discovery rate (FDR) to select the top five with the smallest adjusted *p*-values for further investigation. Visualization of these results was performed using the ggplot2 package (v3.4.2) in R for representation and interpretation of the data.

The ClueGO app (v2.5.9) within Cytoscape (v3.9.1) was utilized to construct a network of Gene Ontology terms for the 89 differentially expressed genes (DEGs) identified in urothelial carcinoma with default settings. This network included terms from the Gene Ontology biological process (GO:BP), Gene Ontology molecular function (GO:MF), and Gene Ontology cellular component (GO:CC) categories.

Gene set enrichment analysis (GSEA) was conducted, stratified by *RHOJ*, utilizing the clusterProfiler package’s GSEA() function (v4.8.1), with parameters set to their defaults [54,55]. The normalized enrichment score (NES) for each gene set was visualized using the ggbarplot() function in the ggpubr package (v0.6.0). The gene sets employed for this analysis were derived from the Hallmark collection of the Molecular Signatures Database (MsigDB), specifically using the ‘h.all.v2022.1.Hs.symbols.gmt’ dataset [56].

### 4.5. ImSig and TIDE Singscore Calculation

To calculate the immune cell infiltration in urothelial carcinoma, we used the imsig package (v1.1.3) in R. The required clinical datasets were downloaded from the TCGA and GSE31684 databases as previously described. The imsig package was installed and loaded in R using the commands install.packages(“imsig”) and library(imsig). The imsig () function from the imsig package was employed to calculate the immune cell infiltration scores for each sample in the TCGA and GSE31684 datasets. This function estimates the abundance of various immune cell types based on the expression profiles of their respective signature genes. The immune infiltration scores were normalized across the datasets to ensure comparability. This was accomplished by subtracting the mean and dividing by the standard deviation for each score. The normalized scores were then used in downstream analyses, including the correlation analysis with *RHOJ* expression. To examine the correlation between *RHOJ* expression and immune cell infiltration in urothelial carcinoma using clinical data from TCGA and GSE31684, we used a correlation matrix analysis in R. The downloaded data were pre-processed to filter out samples that lack information about *RHOJ* expression or immune cell infiltration. The *RHOJ* expression values and immune cell infiltration scores were log-transformed to ensure normal distribution. A correlation matrix was constructed in R using the cor() function, which computes the correlation between *RHOJ* expression and immune cell infiltration levels. This was conducted separately for the TCGA and GSE31684 datasets. The correlation matrix was visualized using the corrplot() function from the corrplot package (v0.92) in R. This function generates a correlation plot, providing an intuitive way of representing the correlation data. We used the cor.test() function in R to calculate *p*-values and assess the statistical significance of the correlations observed. Any correlations with a ***p***-value less than 0.05 were considered statistically significant. For the calculation of TIDE singscore, we adopted the T cell dysfunction signature as published by Peng et al., which comprises immune checkpoint blockade resistance (ICB resist), myeloid-derived stromal cell (MDSC), T cell accumulation (T accu), T cell exhaustion (T exhaust), and regulatory T cell (Treg) signatures [34]. These were divided into two gene sets—upregulated and downregulated. The singscore package (v1.18.0) was used to import two gene sets for each signature, which were then applied to the TCGA BLCA, GSE31684, and GSE32894 datasets for TIDE singscore calculation. Following this, a correlation analysis was conducted with the *RHOJ* expression against the calculated TIDE singscore.

## 5. Conclusions

In summary, our investigation comprehensively examined the implications of *RHOJ* in bladder cancer. We identified its potential application as a prognostic marker, a key participant in disease progression and immune modulation, and a possible therapeutic target. Our findings underscore the need for further research on *RHOJ* to fully appreciate its potential in improving the diagnosis, prognosis, and treatment of bladder cancer.

## Figures and Tables

**Figure 1 ijms-24-14081-f001:**
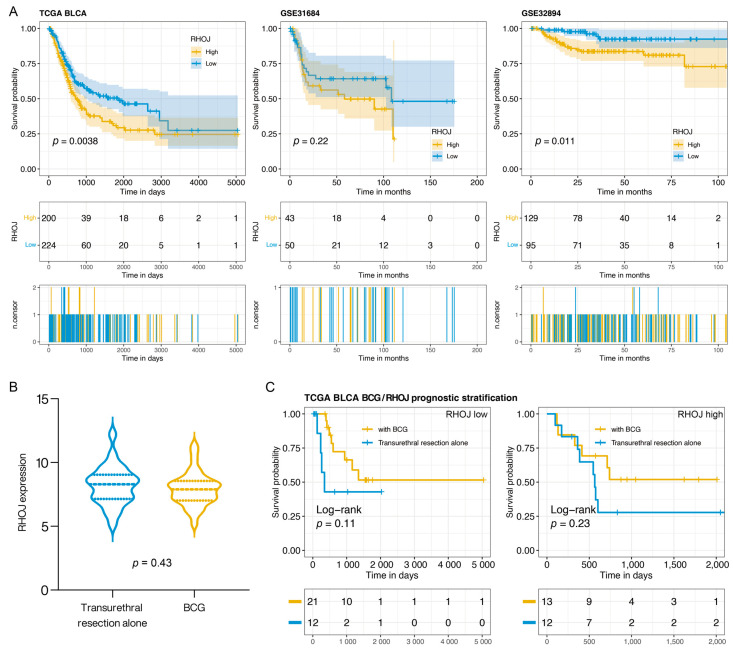
The relationship between *RHOJ* expression and patient prognosis in urothelial tumor. (**A**) Kaplan–Meier survival analysis was performed to evaluate the effect of *RHOJ* gene expression levels on the survival rates of patients with urothelial tumors. The analysis was carried out using three different databases: TCGA BLCA (left panel), GSE31684 (middle panel), and GSE32894 (right panel). Patients demonstrating high *RHOJ* expression (depicted by the yellow curve) showed significantly lower survival rates in comparison to those exhibiting low *RHOJ* expression (illustrated by the blue curve). The differences between these two patient groups were statistically significant as determined by log-rank test. (**B**) Violin plot illustrating the distribution of *RHOJ* expression in bladder cancer patients, stratified by those who received BCG treatment and those who underwent transurethral resection alone. Statistical significance was assessed using the Wilcoxon *t*-test. (**C**) Kaplan–Meier survival plot comparing the prognosis of bladder cancer patients categorized based on *RHOJ* expression levels and treatment type. The yellow curve represents patients who received BCG treatment, while the blue curve signifies those who underwent transurethral resection alone. Survival differences were evaluated using the log-rank test.

**Figure 2 ijms-24-14081-f002:**
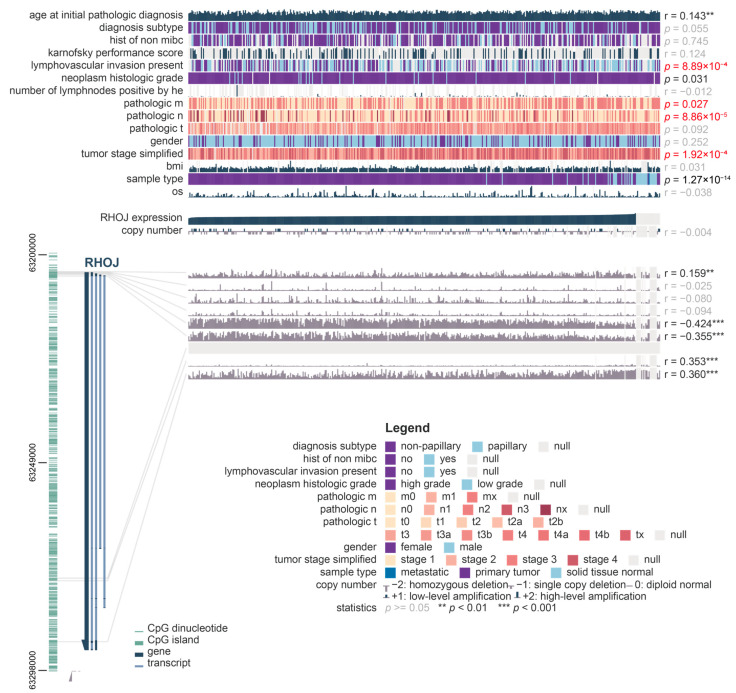
Correlation between *RHOJ* gene expression, methylation levels, and bladder cancer metastasis along with clinical features. MEXPRESS analysis was used on TCGA BLCA database to derive this correlation. The relationships between *RHOJ* expression, methylation level, and clinical factors such as age, gender, survival status, pathological stage, TNM stage, and lympho-vascular invasion were examined. Red text represents indicators associated with clinical tumor metastasis. The comparison was specifically focused on *RHOJ* expression, with Pearson correlation coefficients calculated for each comparison and *p*-values obtained. The significance levels are denoted as **: *p* < 0.01, ***: *p* < 0.001.

**Figure 3 ijms-24-14081-f003:**
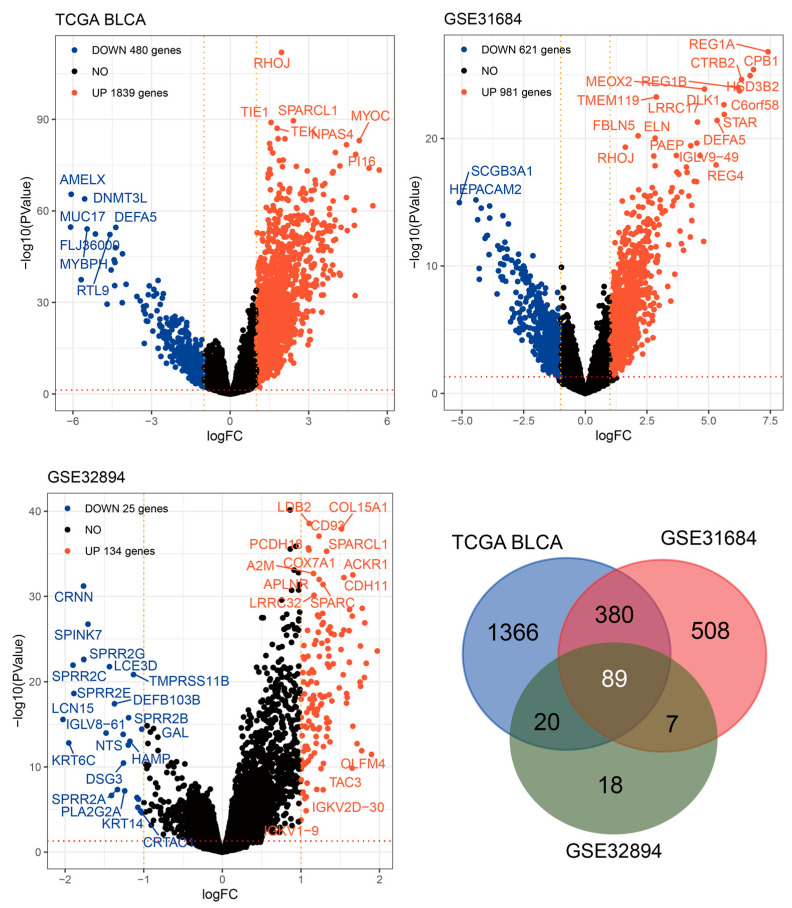
Intersect analysis of *RHOJ*-associated upregulated genes across three databases. Volcano plots were constructed to analyze the gene expression data in bladder cancer patients with high *RHOJ* expression. In these plots, the -log10 false discovery rate (FDR) is plotted against log2 fold change (FC), highlighting significantly upregulated (red) and downregulated (blue) genes. A cutoff of absolute log2FC > 1 and FDR < 0.05 was used to determine significant differential expression. A total of 89 genes were identified as upregulated in association with high expression of *RHOJ*. Please refer to Appendix A for detailed information.

**Figure 4 ijms-24-14081-f004:**
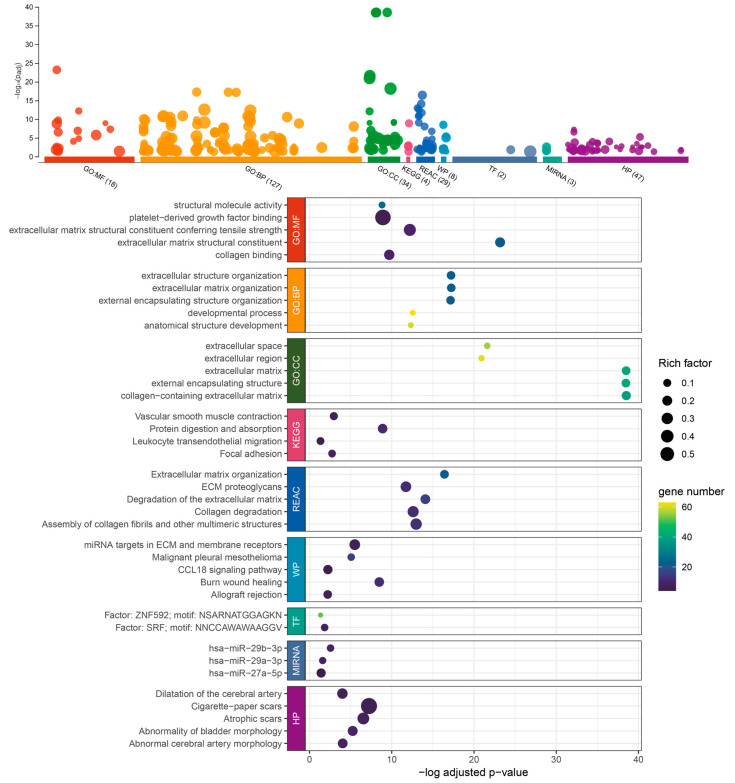
Enrichment analysis of the 89 upregulated genes associated with *RHOJ* in TCGA BLCA, GSE31684, and GSE32894 databases, performed using g:Profiler. g:Profiler was used to analyze the association and enrichment of the 89 upregulated genes in the TCGA BLCA, GSE31684, and GSE32894 databases. The results showed that the majority of these genes were significantly associated with Gene Ontology: molecular function (GO:MF), biological process (GO:BP), cellular component (GO:CC), Reactome pathways (REAC), WikiPathways (WP), transcriptional factor (TF), microRNA (MIRNA), and the link of the upregulated genes to certain disease phenotypes (Human Phenotype Ontology, HP). The x-axis shows the -log of the adjusted *p*-value for each category or pathway’s enrichment. This measure indicates the statistical significance of each enrichment, with higher values (toward the right) denoting higher significance after controlling for false discovery rate due to multiple testing. The size of each point (Rich Factor) represents the ratio of the number of enriched genes in each functional category or pathway to the total number of genes in that category or pathway. The color intensity of each point, depicted using a viridis color scale, represents the number of genes from the study associated with each category or pathway. Colors closer to yellow denote categories or pathways that encompass more genes from the study. Please refer to Appendix A for detailed information.

**Figure 5 ijms-24-14081-f005:**
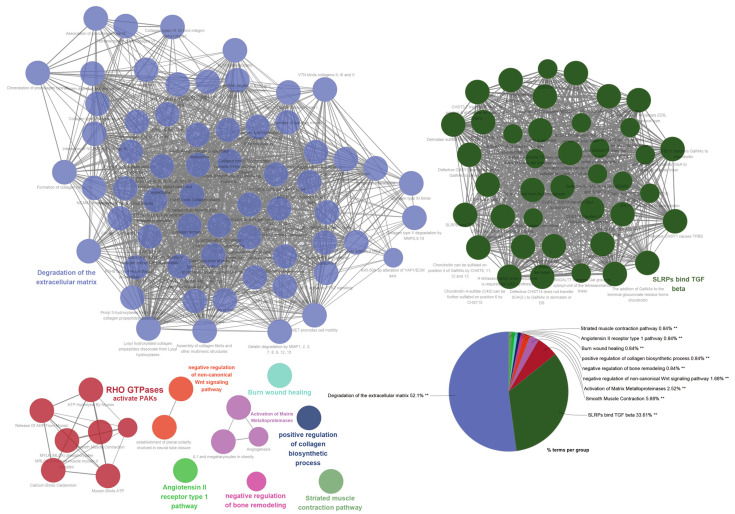
Cluster analysis and functional assessment of the 89 genes associated with *RHOJ*. Cytoscape was used to analyze the functional associations among the 89 upregulated genes in TCGA BLCA, GSE31684, and GSE32894 databases. Each node represents a gene set, and connections (edges) between nodes show functional associations between the gene sets. Gene sets that share more connections tend to have more similar functions. The size of a node typically reflects its relative importance within the network. Nodes are color-coded to represent different functional groups or clusters. Each cluster of tightly connected gene sets represents a group of gene sets involved in a specific biological function. The pie chart provides a summarized view of the Gene Ontology (GO) terms and pathways represented. Each slice of the pie represents a different GO term or pathway. The size of the slice corresponds to the relative frequency. The percentage indicates the proportion of genes associated with the corresponding term. Statistical significance of enriched GO terms is indicated by a double asterisk (**) at a statistical level of *p* < 0.01.

**Figure 6 ijms-24-14081-f006:**
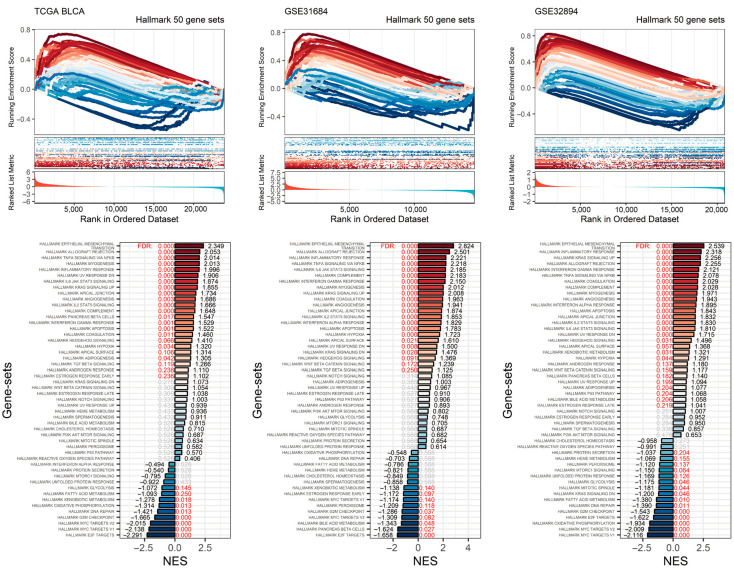
Impact of elevated *RHOJ* expression on Hallmark gene sets in urothelial tumors. The GSEA plot illustrates the enrichment trajectory of Hallmark gene sets. Lines transitioning from red to blue depict the enrichment scores, which are organized in a descending order based on the normalized enrichment score (NES). These scores are labeled at the ends of the NES bars. Red text is used to indicate the false discovery rate (FDR) values, signifying an FDR less than 0.25. Meanwhile, gray text is used to denote instances where there is no statistical significance.

**Figure 7 ijms-24-14081-f007:**
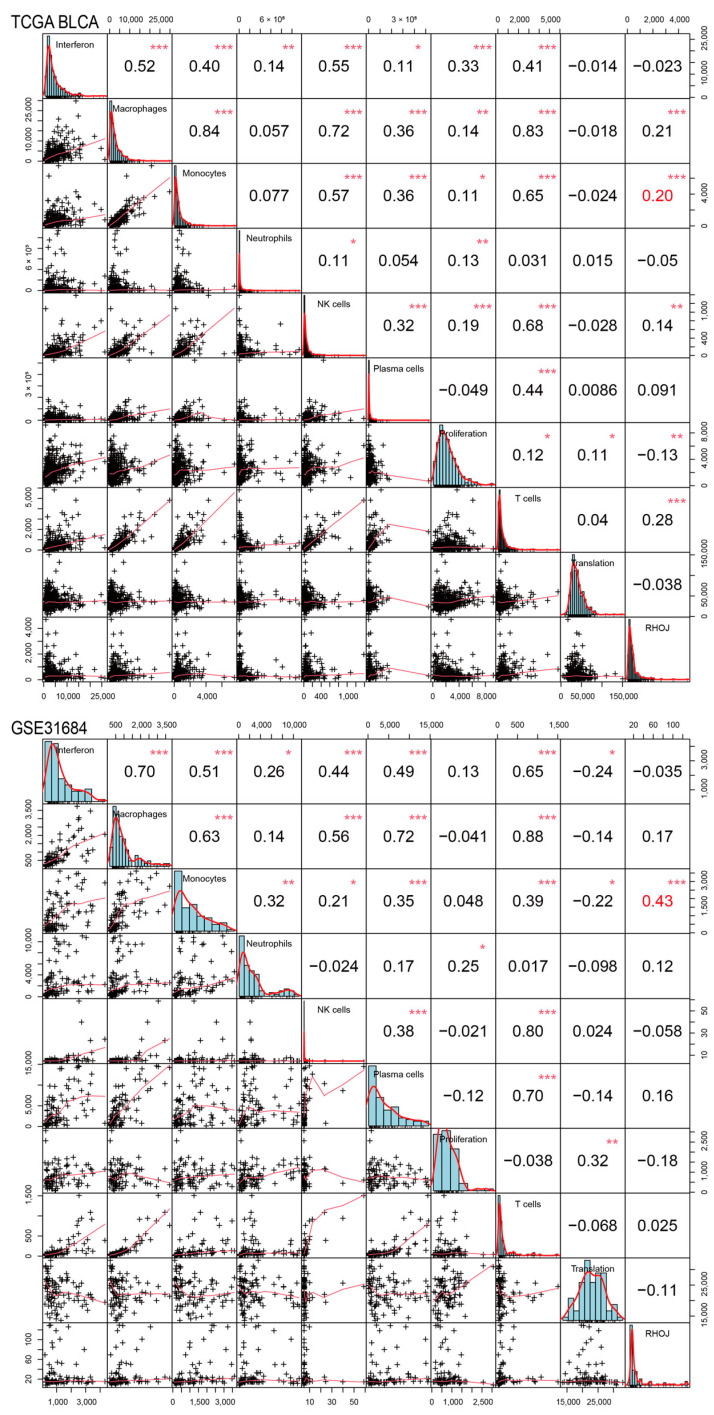
Correlation of *RHOJ* expression with alterations in the ImSig score in patients with urothelial tumors. ImSig analysis was performed on TCGA BLCA (figure above) and GSE31684 datasets (figure below) to assess the association between *RHOJ* expression and various immune cell populations in the tumor microenvironment. The results were presented using Pearson correlation coefficients. Statistical significance is denoted as follows: * *p* < 0.05, ** *p* < 0.01, *** *p* < 0.001. NK cell stands for natural killer cell.

**Figure 8 ijms-24-14081-f008:**
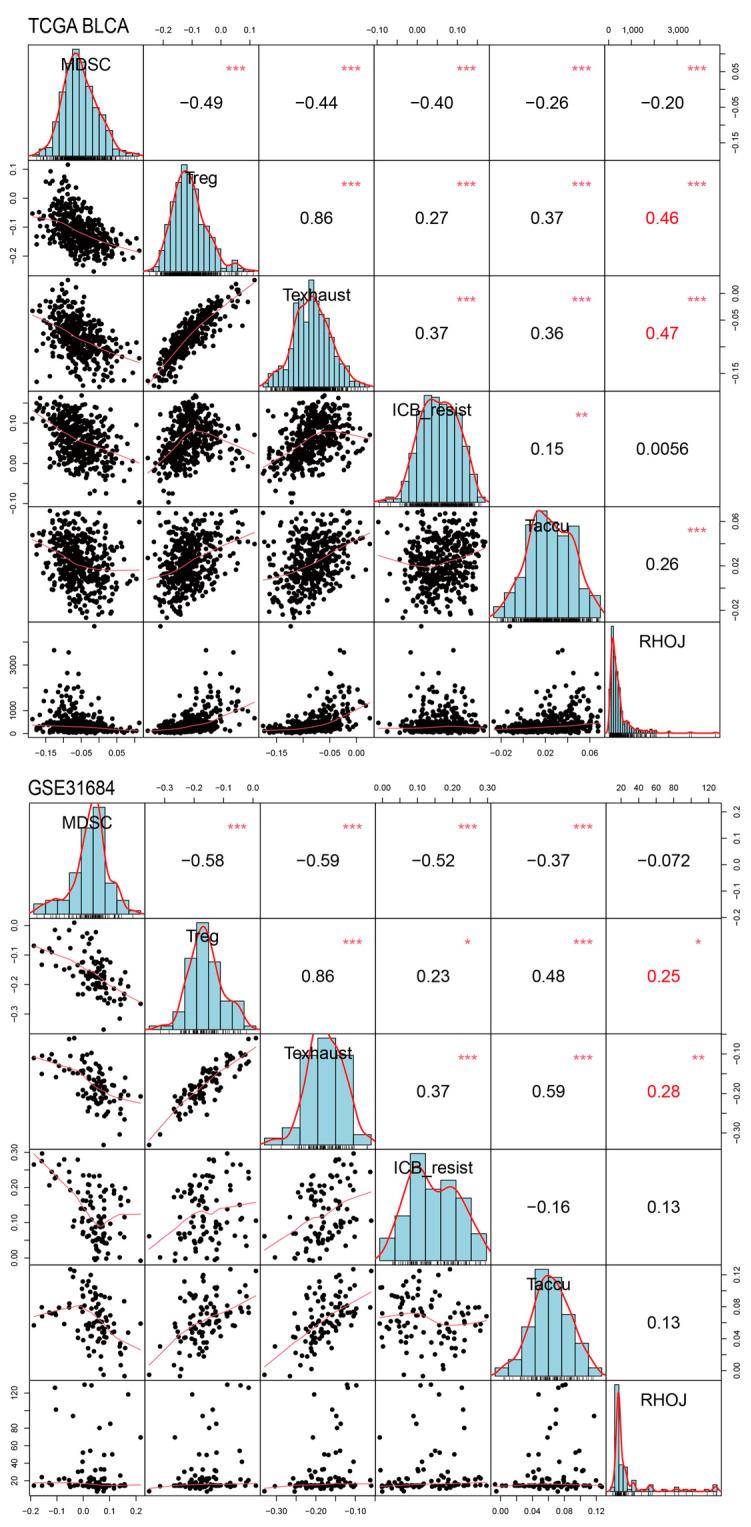
Correlation of *RHOJ* expression with alterations in the Tumor Immune Dysfunction and Exclusion (TIDE) singscores in patients with urothelial tumors. Singscore calculation of TIDE signature was performed on TCGA BLCA (figure above) and GSE31684 datasets (figure below) to assess the association between *RHOJ* expression and suppressive immune cell signatures in the tumor microenvironment. The results were presented using Pearson correlation coefficients. Statistical significance is denoted as follows: * *p* < 0.05, ** *p* < 0.01, *** *p* < 0.001. MDSC: myeloid-derived suppressor cells, Treg: regulatory T cells, Texhaust: T cell exhaustion, ICB resist: immune checkpoint blockade resistance, Taccu: T cell accumulation.

## Data Availability

Publicly available datasets were analyzed in this study. These data can be found here: GSE31684 (https://www.ncbi.nlm.nih.gov/geo/query/acc.cgi?acc=GSE31684 (acessed on 16 November 2022)); GSE32894 (https://www.ncbi.nlm.nih.gov/geo/query/acc.cgi?acc=GSE32894 (accessed on 16 November 2022)).

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
