# Peer review of "Elucidating the Associated Biological Function and Clinical Significance of RHOJ Expression in Urothelial Carcinoma"

_ijms, 2023, doi:10.3390/ijms241814081_

Round 1
Reviewer 1 Report
It is a well designed study which provide provocative information on the significance of RHOJ expression in urothelial cancer.
The experimental design is clear.
The paper should provide additional information in the specific field which can be valuable for the scientific community
The only point that can be described at this point as a perspective for future work is to detail the steps that are now needed for the formal status of biomarker that means the set up for investigation in the clinical setting, taking into account that the described findings are proof on concept as derived by public databases and do not provide a translatable tool for urothelial cancer patients. The authors are just needed to describe these points and hypothesize the scenario for clinical use of the biomarker. The paper includes novel information and merits publication with some efforts to improve writing. With the suggested points a larger audience wil take benefit of information included in the work.A nice and well written paper
Needs some improvement in writing
Author Response
Dear Reviewer,
We deeply appreciate your suggestions and support. Your feedback has been invaluable to our work. Please refer to the attached file for a detailed response.
Warm regards,
Yilin

Reviewer 2 Report
Very well structured work on interesting perspective of bladder carcinogenesis and potential therapeutical target.
It needs just some argumentations to have more clinical appeal too:
- can you add informations about expression on BCG status/ failed patients on datasets?
- any result from upper urinary tract carcinomas in data sets?
Your findings on inflammatory background are very interesting and make a link with multiple clinical investigations about monocytes/lynphocites status (local and systemic) tumoral infiltration and tumoral microenvironment development.
Hiscaemia and neoangiogenesis are however well known paths in urological carcinogenesis.
You could add some point on discussion about this topic.
Finally is very interesting the association of HPO and bladder morphological abnormalities, this is another food for thought as congenital abnormalities are often a driver for other urological neoplasms and for bladder carcinogenesis too:
you can cite for example:
Murugapoopathy V, Gupta IR. A Primer on Congenital Anomalies of the Kidneys and Urinary Tracts (CAKUT). Clin J Am Soc Nephrol. 2020 May 7;15(5):723-731. doi: 10.2215/CJN.12581019. Epub 2020 Mar 18. PMID: 32188635; PMCID: PMC7269211.
Leoni C, Paradiso FV, Foschi N, Tedesco M, Pierconti F, Silvaroli S, Diego MD, Birritella L, Pantaleoni F, Rendeli C, Onesimo R, Viscogliosi G, Bassi P, Nanni L, Genuardi M, Tartaglia M, Zampino G. Prevalence of bladder cancer in Costello syndrome: New insights to drive clinical decision-making. Clin Genet. 2022 Apr;101(4):454-458. doi: 10.1111/cge.14111. Epub 2022 Feb 17. PMID: 35038173.
Author Response
Dear Reviewer,
We deeply appreciate your suggestions and support. Your feedback has been invaluable to our work. Please refer to the attached file for a detailed response.
Warm regards,
